# Immune Checkpoint Inhibitors in Hepatocellular Carcinoma and Their Hepatic-Related Side Effects: A Review

**DOI:** 10.3390/cancers16112042

**Published:** 2024-05-28

**Authors:** Thomas M. Ruli, Ethan D. Pollack, Atul Lodh, Charles D. Evers, Christopher A. Price, Mohamed Shoreibah

**Affiliations:** 1Internal Medicine Residency Program, Department of Medicine, University of Alabama at Birmingham, Birmingham, AL 35294, USA; edpollack@uabmc.edu (E.D.P.); alodh@uabmc.edu (A.L.); christopherprice@uabmc.edu (C.A.P.); 2Division of Gastroenterology and Hepatology, Department of Medicine, University of Alabama at Birmingham, Birmingham, AL 35233, USA; mshoreibah@uabmc.edu

**Keywords:** hepatocellular carcinoma, PD-1, PD-L1, CTLA-4, immunotherapy, immune checkpoint inhibitors, hepatotoxicity, hepatitis

## Abstract

**Simple Summary:**

Immunotherapy has evolved as a leading therapeutic modality for several malignancies, and increasing evidence has shown its utility in the treatment of hepatocellular carcinoma. Specifically, immune checkpoint inhibitors have shown promise as an alternative treatment for this neoplasm. However, given their ability to reactivate the immune system, these treatments pose significant risks, specifically hepatitis viral reactivation, autoimmune hepatitis, and hepatotoxicity. Here, we aim to review the potential hepatic side effects associated with immune checkpoint inhibitors in the treatment of hepatocellular carcinoma. In doing so, we hope to provide an effective summary detailing our current understanding of the true risk of developing these side effects for providers considering immune checkpoint inhibitor therapy for their patients with hepatocellular carcinoma.

**Abstract:**

Primary liver cancer is one of the leading causes of cancer mortality worldwide, with hepatocellular carcinoma (HCC) being the most prevalent type of liver cancer. The prognosis of patients with advanced, unresectable HCC has historically been poor. However, with the emergence of immunotherapy, specifically immune checkpoint inhibitors (ICIs), there is reason for optimism. Nevertheless, ICIs do not come without risk, especially when administered in patients with HCC, given their potential underlying poor hepatic reserve. Given their novelty in the management of HCC, there are few studies to date specifically investigating ICI-related side effects on the liver in patients with underlying HCC. This review will serve as a guide for clinicians on ICIs’ role in the management of HCC and their potential side effect profile. There will be a discussion on ICI-related hepatotoxicity, the potential for hepatitis B and C reactivation with ICI use, the potential for the development of autoimmune hepatitis with ICI use, and the risk of gastrointestinal bleeding with ICI use. As ICIs become more commonplace as a treatment option in patients with advanced HCC, it is imperative that clinicians not only understand the mechanism of action of such agents but also understand and are able to identify hepatic-related side effects.

## 1. Introduction

As of 2020, primary liver cancer was the sixth most common cancer and the third leading cause of cancer death worldwide. Of the primary liver cancers, hepatocellular carcinoma (HCC) is the most prevalent type, accounting for approximately three-fourths of total cases [1]. The incidence of HCC varies across the globe, and incidence rates have been found to be decreasing in known “high-rate areas” (Asia and Africa) but increasing in previously known “low-rate areas” (the Americas, India, most European countries) [2]. Unfortunately, the prognosis of patients with advanced HCC is relatively poor. For example, the median survival following diagnosis is approximately six to twenty months [3].

Historically, the treatment options for HCC have included surgical resection of the tumor, ablative therapies, loco-regional therapies (such as trans-arterial chemoembolization [TACE] or radiofrequency ablation), targeted systemic chemotherapy, and liver transplantation. The management largely depended on patients’ performance status, the reserve of liver function, and tumor staging [4,5].

Until 2017, the only systemic therapies used in the advanced stages of HCC included anti-angiogenic tyrosine kinase inhibitors (TKIs), such as sorafenib, lenvatinib, and regorafenib, to name a few [6]. What has emerged recently is the role of immunotherapy in the treatment of HCC, specifically immune checkpoint inhibitors (ICIs). ICIs are humanized monoclonal antibodies targeting particular immune checkpoint proteins. They have revolutionized cancer treatment for a variety of cancers, including metastatic melanoma, head and neck cancers, non-small cell lung cancer, renal cell carcinoma, and breast cancer (to name a few), and their role in the treatment of HCC is an exciting one [6,7,8,9].

Although ICIs are showing promise in the treatment of HCC, they do not come without risk. Nearly every organ system can experience an ICI-related adverse event, and clinicians must be vigilant to recognize these events as they can be severe or even life-threatening. Examples of ICI-related adverse events include, but are not limited to, encephalitis, pneumonitis, myocarditis, rash, colitis, and glomerulonephritis. Another example of an ICI-related adverse event—and a major topic of this paper—will be ICI-related hepatotoxicity/hepatitis [10]. This adverse event can become a tricky one in patients with HCC since their liver function is likely already affected by a baseline diseased liver [11].

## 2. Immune Checkpoint Inhibitors and Their Role in Treating HCC

Most recently, as our understanding of HCC (and malignancies in general) has grown, so have the options to treat and manage it. Indeed, immunotherapy has evolved as a more novel approach to treating many types of malignancies, including HCC [12,13,14]. Immunotherapies that have emerged include adoptive cell therapy, cancer vaccines, cytokine therapy, and immune checkpoint inhibitors (ICIs), among others [15]. The immune system is a powerful tool that can be used in the treatment of a malignancy. Both innate and adaptive immune systems play a role in both the prevention of cancer and in preventing cancer progression [6,16]. Interestingly, while the immune system has been shown to play a role in protecting the body against cancer or tumor formation, it can also play a role in cancer growth and spread. It specifically plays this role in malignancies that have created mechanisms to escape immune recognition, such as HCC [17]. In order to understand the role of ICIs in their treatment of HCC, there will need to be a discussion on HCC’s tumor microenvironment and ICIs’ mechanisms of action.

At its baseline, the liver exists in an “anti-inflammatory” state; this is necessary, given its near constant exposure to foreign antigens (such as food molecules) [6]. Kupffer cells (KCs), hepatic stellate cells (HSCs), and liver sinusoidal endothelial cells (LSECs) are all examples of non-parenchymal hepatic cells that help contribute to an “anti-inflammatory” state of the liver. KCs are the “resident liver macrophages” that, along with HSCs and LSECs, act as antigen-presenting cells (APCs) in the liver. All three of these cell types can present antigens to T cells [18]. KCs have been shown to produce molecules (such as IL-10 and prostaglandins) that activate regulatory T cells; HSCs release a molecule called the hepatocyte growth factor, which contributes to regulatory T cell accumulation, and LSECs express a high level of PD-L1 [6,19,20,21]. Overall, all three cell types have been shown to play a role in the “anti-inflammatory” state of the liver.

HCC is an example of a malignancy that has developed ways to evade immune recognition. One of the ways that it accomplishes this is by producing what is known as an immunosuppressive tumor microenvironment (TME). An immunosuppressive TME allows for decreased recognition of tumor cells by cells of the immune system [6,16]. The immunosuppressive TME of HCC is complex and made up of a variety of hepatic non-parenchymal cells (KCs, HSCs, LSECs), immune cells, tumor cells, and tumor-associated fibroblasts [6]. The combination of and interaction between these different types of cells drives the generation of the immunosuppressive TME seen in HCC and can lead to what is known as “T-cell exhaustion”, which has been found to be associated with deleterious effects on the liver. The interactions between these cells can lead to an increase in the number of cells that provide negative regulatory immune activity (such as inhibitory B cells, regulatory T cells), elevated levels of tolerogenic enzymes, reduced immunoglobulin-assisted opsonization, and they can also lead to the upregulation of co-inhibitory lymphocyte signals which include immune checkpoint receptors and ligands [6,22]. The latter mechanism is the target of ICIs. Examples of these immune checkpoint receptors and ligands include cytotoxic T lymphocyte-associated antigen 4 (CTLA-4; also known as CD152) and programmed cell death protein 1 (PD-1; also known as CD279) [6,23].

CTLA-4 is a molecule expressed by regulatory T cells. The CTLA-4 molecule competes with CD28 in binding to its ligands, CD80 and CD86, which are both typically found on antigen-presenting cells. The binding of CD80/86 to CD28 results in a co-stimulatory immune response (amplifying the immune response), while the binding of CTLA-4 to CD80/86 results in a co-inhibitory response (dampening the immune response). CTLA-4 binding to CD80/86 inhibits CD28 signaling and, thus, plays a role in downregulating immune responses and allowing for “self-tolerance” [6,23,24]. This downregulation of the immune response via the CTLA-4 pathway inhibits the activation of T cells and can lead to a decrease in their ability to recognize tumor cells and antigens. During a normal immune response, the CTLA-4 pathway helps to prevent overactive T cell immune responses and assists in halting activated T cell activity when their action is no longer necessary in an immune mediated response. This CTLA-4 pathway plays a vital role in normal physiologic conditions. However, in the setting of a malignancy, this pathway can play a role in preventing the activation of activated T cells that ultimately assist in recognizing cancer antigens. Furthermore, in the immunosuppressive TME, CTLA-4 amplifies regulatory T cells and impedes the function of APCs, such as dendritic cells [9]. Importantly, CTLA-4 has been found to be overexpressed in certain types of HCC, which can lead to the uncontrolled growth and spread of the tumor via this “downregulating” pathway [9,25,26]. Administering therapy with anti-CTLA-4 monoclonal antibodies may function as a way to overcome this immunosuppressive pathway and allow for an increase in the immune recognition of the malignancy [9].

PD-1 is expressed on a wider variety of cells in comparison to CTLA-4; it can be found on natural killer (NK) cells, activated T cells, B cells, macrophages, monocytes, and dendritic cells. Furthermore, PD-1 has been noted to be highly expressed on tumor-specific T cells in some cases [27,28]. PD-1 ligand (PD-L1) can be found to be expressed on macrophages, activated B and T cells, and dendritic cells. Furthermore, and perhaps most importantly, they can be found to be expressed by tumor cells as a way to avoid immune-mediated anti-tumor responses [27,28,29]. When PD-L1 binds to PD-1, a number of downstream effects occur. Firstly, apoptosis is induced in antigen-specific T cells; secondly, apoptosis is reduced in regulatory T cells. The result of this leads to the inhibition of immune effector functions, helps to maintain immune tolerance, and reduces pro-inflammatory immune responses. The binding of PD-1 to PD-L1 essentially causes T-cell dysfunction and neutralization. This “dampening” of the immune response may lead to the progression of cancer growth and spread by inhibiting what would be a protective, proper, anti-tumor immune response [6,9,27]. Administering anti-PD-1 monoclonal antibodies can inhibit this dampening escape mechanism which, in turn, allows for T cells to remain in an “active state” and to exert their inherent anti-tumor/anti-cancer response. The pathway affected by anti-PD-L1 monoclonal antibodies is similar to that of anti-PD-1 monoclonal antibodies; the only differing factor is the target of the antibody [9].

This targeted pathway from PD-1/PD-L1 is an important one in the field of immunotherapy. PD-L1 has been shown to be expressed in various cancers—including gastric cancer, esophageal cancer, renal cell carcinoma, ovarian cancer, and hepatocellular carcinoma—and the overexpression of this immunomodulating ligand has been associated with poorer clinical outcomes. Furthermore, in a study by Calderaro et al., who investigated a large series of patients with HCC, PD-L1 was expressed in 17% of the tumors [30]. The overexpression of this molecule by tumor cells plays a role in preventing the host immune response from neutralizing it, leading to immune escape [9,31]. The ability to stymie this immune escape mechanism via anti-PD-1/anti-PD-L1 antibodies (a type of ICI) makes it an attractive target.

## 3. Examples of Immune Checkpoint Inhibitors in HCC Treatment

Two examples of anti-CTLA-4 monoclonal antibodies currently approved for use in HCC treatment are tremelimumab and ipilimumab.

Tremelimumab was first used to evaluate both the anti-tumor and antiviral effects of the CTLA-4 blockade in patients with HCC and chronic hepatitis C viral (HCV) infection in a clinical trial carried out in 2013. In the trial, among the 17 patients that were evaluated for HCC tumor response, the overall response rate (ORR) was 17.6% [32]. Early results inspired further studies to investigate CTLA-4 inhibitors’ role in HCC treatment. In a 2017 study, tremelimumab, in combination with durvalumab (an anti-PD-L1 immune checkpoint inhibitor), was investigated in patients with unresectable HCC; it was a phase I/II safety and efficacy clinical trial. In total, 40 patients were enrolled in the study; 100% of patients had a partial response (PR) and 15% of patients had an ORR [33]. In 2020, an expansion of this study was presented, and it included part three of phase II. This part of the study investigated several different treatment regimens (durvalumab and tremelimumab monotherapy, as well as different doses of combination therapy) and included 332 patients. The results from this part of the study revealed that the “higher dose” arm of the combination of tremelimumab and durvalumab resulted in an ORR of 24% and a medial overall survival of 18.7 months (both the highest values in comparison to the other arms of the study) [34]. Given the encouraging results from the phase II trial, a phase III clinical trial was designed. This trial is known as the HIMALAYA trial [35]. The phase III trial compared durvalumab monotherapy, the “higher dose” regimen from the previous study (“STRIDE”), and sorafenib (a protein kinase inhibitor that was considered ‘standard of care’ for patients with unresectable HCC for over a decade) [35,36]. In total, 1171 patients with unresectable HCC were randomly assigned to receive durvalumab monotherapy, “STRIDE”, or sorafenib. The STRIDE regimen was found to significantly improve overall survival versus the previously considered ‘standard of care’ sorafenib (overall survival at 36 months was 30.7% vs. 20.2% in STRIDE and sorafenib, respectively) [35].

Ipilimumab was first evaluated for use in HCC treatment in combination with nivolumab (an anti-PD-1 immune checkpoint inhibitor) in cohort 4 of the CheckMate040 trial [37]. The objective of this trial was to assess the safety and efficacy of both ICIs in patients with advanced HCC in a sub-cohort of patients who were previously treated with sorafenib. In the randomized trial, the nivolumab–ipilimumab combination regimen had good safety profiles, promising overall survival rates, and encouraging durable response rates. Because of these promising results, the nivolumab–ipilimumab combination received accelerated approval in the United States (as a second-line therapy for HCC), and further studies were initiated to evaluate the possibility of this combination as a first-line therapy for patients with HCC [37]. In the CheckMate 9DW trial, this combination is being compared to sorafenib or lenvatinib as a first-line therapy for patients with advanced HCC; its estimated study completion date is June of 2025 [9,38].

Two examples of anti-PD-1 monoclonal antibodies currently approved for use in HCC treatment are nivolumab and pembrolizumab.

Nivolumab was the first ICI approved for the treatment of HCC [39]. This ICI’s role in the treatment of HCC was first investigated in a multicenter, phase I/II, open-label trial known as the CheckMate040 trial, where the safety and efficacy of nivolumab in patients with advanced HCC was investigated [40]. The results of the study were promising, resulting in an accelerated approval of nivolumab by the FDA in 2017 and resulting in another trial, known as the CheckMate459 trial, which investigated nivolumab as a potential superior therapy compared to sorafenib (the at-the-time first-line therapy) [9,41]. This phase III trial failed to show superiority of nivolumab as a first-line therapy compared to sorafenib, and further analyses of nivolumab’s role in HCC treatment are needed [9,39,41]. As mentioned previously, nivolumab has been studied in combination with ipilimumab for patients with advanced HCC.

Pembrolizumab is another example of an anti-PD-1 monoclonal antibody used in the treatment of HCC [9,39]. Pembrolizumab was approved as a second-line systemic therapy for patients with advanced HCC as a result of the Keynote224 trial [42]. The phase II trial found that pembrolizumab was both safe and efficacious in patients with advanced HCC (who had previously been treated with sorafenib); as a result, a randomized, double-blind phase III trial—known as Keynote240—was designed to compare pembrolizumab versus “best supportive care” versus “placebo plus best supportive care” [43]. Although there was an improvement in overall survival and progression-free survival in the pembrolizumab group compared to the placebo, they did not reach statistical significance per the pre-specified trial criteria [43]. However, despite this, the authors concluded that there is a favorable risk to benefit ratio for pembrolizumab in patients with advanced HCC. Most recently, in 2023, the Keynote394 trial (a double-blind, phase III trial) evaluated the efficacy and safety of pembrolizumab versus a placebo in patients from Asia with advanced HCC (who were previously treated) [44]. The results from this study demonstrated that pembrolizumab significantly prolonged overall survival and progression-free survival compared to the placebo [44].

Two examples of anti-PD-L1 monoclonal antibodies currently approved for use in HCC treatment are atezolizumab and durvalumab.

Atezolizumab is an anti-PD-L1 monoclonal antibody that is used in the treatment of HCC; it was the first anti-PD-L1 antibody approved by the FDA for the treatment of a number of cancers [9]. The use of atezolizumab in HCC treatment is most recognized in combination therapy, specifically in the setting of combining this therapy with the anti-vascular endothelial growth factor (VEGF) antibody, bevacizumab. The combination of these two agents was first studied in an open-label, phase Ib study (known as GO30140) where atezolizumab with or without bevacizumab was studied in patients with unresectable HCC [45]. This study demonstrated a longer progression-free survival in the combination therapy group compared to atezolizumab monotherapy [45]. Because of this successful trial, a phase III, open-label trial (known as the IMBrave150 trial) was designed to evaluate the combination of atezolizumab with bevacizumab compared to standard-of-care sorafenib in patients with unresectable, advanced HCC [46]. This study found that in this patient population, a combination of atezolizumab with bevacizumab resulted in better overall survival and progression-free survival compared to patients who received sorafenib [46].

Another example of an anti-PD-L1 antibody used in HCC treatment is durvalumab [9]. Durvalumab was investigated in a phase I/II study to evaluate the efficacy and safety of the ICI in patients with advanced solid tumors. Upon the evaluation of the HCC sub-cohort, durvalumab showed promising results in both anti-tumor activity and overall survival [47]. Similar to atezolizumab, durvalumab is being investigated in combination with bevacizumab in the Emerald-2 trial; its estimated completion date is 2025 [9,48]. The Emerald-1 trial is a study currently underway that is investigating trans-arterial chemoembolization (TACE) in combination with durvalumab and bevacizumab in patients with loco-regional HCC; its estimated completion date is 2024 [49].

Although ICIs have shown benefits and promise in patients with HCC, they do not come without potential adverse events. ICI-related adverse events have been shown to involve nearly every organ system. In patients with hepatocellular carcinoma (HCC), the overall incidence of ICI-related adverse events is not particularly different compared to ICI-related adverse events in patients with other cancer types. However, data have found that the incidence of ICI-related hepatotoxicity/hepatitis is slightly higher in patients with HCC compared to patients with other cancer types [50]. HCC can arise from a number of different etiologies (for example, MAFLD, HBV, HCV, alcoholic liver disease); these different etiologies all lead to a chronic inflammatory state in the liver and may lead to a cirrhotic microenvironment that is the perfect setting for the development and progression of HCC [9,50,51]. Given the fact that many patients with HCC have underlying liver cirrhosis or liver disease, their overall hepatic function may be compromised. Their hepatic reserve may be less than that of a patient with a healthy liver, which puts them at higher risk to develop hepatic related side effects to ICIs [9]. The following section will discuss the potential hepatotoxic effects of ICIs.

## 4. Hepatotoxicity

Hepatotoxicity is a well-established immune-related adverse event (IRAE) associated with the utilization of ICIs. The exact pathogenesis of ICI-induced hepatotoxicity (ICH) has not been fully elucidated, but it is likely multifactorial. ICIs provide therapeutic and survival benefits by augmenting the immune response towards tumor cells, especially in the immunosuppressive TME, as previously described. However, the loss of T-cell inhibition/exhaustion also causes impaired self-tolerance, enhanced inflammatory cascades, and complement activation, ultimately leading to IRAEs. Specifically for the liver, this leads to T-cell-mediated hepatitis, hepatocyte injury, and occasionally cholangitis [52,53]. Patients with HCC have a limited liver reserve and often have cirrhosis with varying degrees of liver dysfunction at baseline [54]. The baseline liver dysfunction seen in these patients can make the identification of ICI-induced hepatotoxicity difficult to diagnose; thus, understanding the epidemiology, clinical presentations, appropriate monitoring, and treatment for hepatotoxicity in patients with HCC cannot be understated.

ICI-induced hepatotoxicity and associated liver dysfunction is classified by two grading systems—Common Terminology Criteria for Adverse Events (CTCAE) and the drug-induced liver injury (DILI) network grading systems. As outlined below in Table 1, both scales consider the degree of liver enzyme elevations and bilirubin derangements. However, the DILI network focuses on additional hepatic decompensation parameters, namely INR, ascites, and signs of encephalopathy [53].

The differences in grading parameters between the CTCAE and DILI network lead to varying degrees of ICH incidence between specific ICI studies. Additionally, ICH varies tremendously depending on the specific ICI, number of agents used, dosing, and the specific malignancy for which treatment is indicated. ICH incidence is reported anywhere from 0 to 30% for all grades and from 1 to 20% for grades 3–4. ICH incidence also increases if higher doses of ICIs are required. The most common timeframe for ICH occurs 8–12 weeks after the initiation of therapy, but it varies by agent ranging from within the first few weeks to multiple months, or even years, after the initiation of therapy. CTLA-4 inhibitors, such as ipilimumab, have historically demonstrated a higher incidence of ICH compared to PD-1 inhibitors, such as nivolumab and pembrolizumab [52,53,55]. However, larger datasets suggest that ICH may be quite similar between CTLA-4 agents and PD-L1 inhibitors, with incidence rates ranging from 1 to 17% overall and 3 to 5% for ICH grades 3–4. The incidence for all ICIs was initially thought to be dose-dependent. Interestingly, newer datasets suggest that, unlike the CTLA-4 inhibitors, there may not be a dose-dependent risk of hepatotoxicity with PD-L1/PD-1 inhibitors [53,55]. When used together for a synergistic effect, ICIs demonstrate higher rates of all IRAEs, including hepatotoxicity. The proportion of hepatotoxicity and other IRAEs has previously been thought to also be a predictor of immunotherapy success and tumor responsiveness. For instance, patients treated with higher doses of ipilimumab for melanoma demonstrated improved survival benefits but had an increased number of cases of IRAEs, including ICH [53,56]. The majority of studies publishing overall incidence were using ICIs in malignancies other than HCC. However, in the IMBrave150 trial, the overall incidence of a grade 3/4 increase in ALT and AST was slightly higher in the atezolizumab–bevacizumab group compared to the sorafenib group [46].

As mentioned above, the pathogenesis of ICH relates to heightened T-cell activity and unregulated immune responses, ultimately leading to an increased ability to recognize evading tumor cells but also organ damage—specifically hepatocyte injury, in this case. The upregulation of the immune system is accomplished via various mechanisms, depending on the ICI. The binding of PD-L1 to its receptor on immune cells leads to their downregulation and protects peripheral tissue from cytotoxic T-cell attacks under physiologic conditions. Monoclonal antibodies directed at PD-1 and PD-L1 block the immune “escape” by tumor cells, leading to a cascade of immunologic activity and tumor cell death [57]. A similar T-cell hyperactivity is accomplished via the modulation of the CTLA-4 receptor. CTLA-4 monoclonal antibodies allow for systemic immune activation and the downstream activation of cytotoxic T-cells, which partly leads to cellular destruction via the release of interferon-gamma, interleukin-2, granzyme, and granulysin [57]. While these mechanisms of immune hyperactivation place patients at risk of ICH, there remains significant heterogeneity in those affected by ICH. Thus, additional factors have been hypothesized to play a role in IRAEs and ICH, including genetic predisposition, underlying autoimmune disease, environmental and drug exposures, and distortion in the gut microbiota [58]. These factors may lower the threshold for patients to present with ICH, given their previous history of triggering events that lead to T-cell activation within the liver.

ICH is predominantly asymptomatic in mild cases and even in some moderate/severe grades of ICH. These presentations may only be identified during interval LFT monitoring while patients are on ICIs. Symptoms associated with ICH include right-sided abdominal pain, fatigue, general malaise, fever, nausea, maculopapular rash, and rarely jaundice [53,59]. Fortunately, presentations of acute liver failure secondary to ICH are rare. Two separate retrospective studies conducted by Gauci et al. and Parlati et al. found no cases of acute liver failure in those treated with ICIs; however, sparse case reports of acute liver failure due to ICH have been reported [60,61,62,63].

Since ICH presents similarly to other liver injuries, a comprehensive evaluation to exclude other forms of liver injury is imperative. Utilizing the Roussel Uclaf Causality Assessment Method (RUCAM) scoring system alongside a thorough history and physical exam is helpful in establishing the diagnosis. Major aspects of the history to be aware of relate to recent drug exposure, herbal supplements, history of autoimmune disease, risk factors for hepatitis B or C (endemic areas, sexual history, IV drug use, tattoos, transfusions), metabolic syndrome (obesity, diabetes, hyperlipidemia—increasing the risk for nonalcoholic fatty liver disease), previous chemotherapy/radiation, history of hypotension, and acetaminophen exposures [64,65]. Utilizing the RUCAM scoring system helps in establishing causality, which can be especially difficult in patients with HCC who often have underlying liver dysfunction. It considers the time to the onset of elevated LFTs after initiating drug treatment, the course of LFT elevations after cessation, alcohol, use, age, concomitant drugs, alternative causes, and responses to drug re-exposure [53,64]. Although rare in the setting of ICIs, important aspects of the physical exam should center around signs concerning liver failure, including asterixis, ascites, caput medusa, hepatomegaly, jaundice, and scleral icterus. Laboratory work-ups and monitoring are necessary for all patients on ICIs as the majority of ICH is asymptomatic. AST, ALT, and alkaline phosphatase are useful in grading the severity of a liver injury. Testing for viral infections, including hepatitis (A, B, and C), other infections (EBV, CMV, HSV, VZV, adenovirus), and toxins (ETOH), and hepatotoxic medications are helpful to rule out other forms of liver injury. Additionally, testing for liver auto-antibodies, including ANA, ASMA, or anti-LMK-1/CYP2D6 antibodies, is recommended. Depending on the clinical presentation, hepatic ultrasound with Dopplers or cross-sectional imaging can be helpful to rule out pathologies, such as malignancy progression or portal vein thrombosis [64,65,66,67,68]. There is no general consensus regarding the necessity of a liver biopsy, but it tends to have more utility as the grade of ICH increases [69]. However, these patients also tend to be at a higher risk for procedures, so this decision should be tailored to the overall clinical picture. Furthermore, in Li et al.’s study (2020), patients receiving a liver biopsy were less likely to receive steroids for a grade 3 or 4 elevation in LFTs and displayed a trend towards a longer time to normalization of LFTs. Notably, patients with HCC were excluded from this study [70].

Although the clinical presentation is similar between autoimmune hepatitis and ICH, there remains no distinct auto-antibodies or immunoglobulins specific for ICH. This is in comparison to AIH, where liver auto-antibodies including antinuclear antibodies, perinuclear anti-neutrophil cytoplasmic antibodies, and anti-smooth muscle antibodies are more commonly seen. Although ICH is a diagnosis of exclusion, the majority of patients with ICH will not require a liver biopsy, and it is not mandatory in the diagnosis [53,71,72]. This is in contrast to AIH, where a liver biopsy is largely utilized to confirm the diagnosis. In a retrospective study of 28 patients, Mar Riverio-Barciela et al. (2020) found that a liver biopsy in patients with ICH was limited to those with a poor or slow response to steroids [73]. For those that do undergo a liver biopsy, there is no pathognomonic finding for ICH. The most common histologic findings show that panlobular inflammation is consistent with acute hepatitis along with the foci of centrilobular necrosis, periportal inflammation, and perivenular infiltrate with endothelialitis [69,71].

Overall, the management of ICH depends largely on its severity. As a general principle, the majority of therapy stems from the discontinuation of the offending ICI and the consideration of immunosuppression with steroids as a first-line therapy for ICH grades 2–4. These details are outlined below according to The American Society of Clinical Oncology guidelines (Table 2). Current recommendations allow for continued ICI therapy in the setting of grade 1 ICH with 1–2 weeks of monitoring. Interestingly, even for ICH grades 3–4, there is some evidence that discontinuation alone results in the normalization of LFTs for ~50% of patients [74]. Additionally, the evidence remains controversial for the therapy’s efficacy at higher doses of steroids (above 60 mg of prednisolone) regarding how much it improves the time to normalization of LFTs [75]. Second-line therapies for ICH that do not respond to steroids include mycophenolate mofetil or azathioprine. Although the frequency of monitoring varies depending on its severity, interval monitoring for the resolution of ICH is essential as “rebound hepatitis” has been documented [76,77,78,79]. In a study involving 28 cases of severe ICH, 9 patients exhibited rebound hepatitis after a premature reduction or cessation of their corticosteroids. Notably, once their corticosteroids were reintroduced, these patients displayed a subsequent improvement in their hepatitis [73]. 

Although controversial, the decision for the re-initiation of ICI treatment also largely depends on the severity of ICH. In general, the expert societies recommend against re-initiating ICIs in patients who suffered from grade III or grade IV ICH. In the case of a grade II injury, guidelines recommend a re-challenge after a temporary cessation of the ICI with a resolution of the liver injury [53,76,77,78,79]. Interestingly, there have been reports of successfully re-starting ICIs in patients with a grade III or IV injury. For example, Riveiro-Barciela et al. investigated re-treatment with ICIs after previous grade III or IV ICI-related hepatitis. Fifteen of the twenty-three patients did not have recurrent cases of ICI-related hepatotoxicity after re-treatment. The majority of these patients received the same ICI during the rechallenge period [80]. Furthermore, Li et al. investigated patients with melanoma who were re-challenged with ICIs after previously developing grade III or IV ICI-related hepatitis. Of the 102 patients who developed grade III or IV ICI-related hepatitis, thirty-one underwent ICI therapy re-challenge. Recurrent ICI-related hepatitis occurred in only four of these patients; however, all of these patients ultimately required ICI discontinuation [70]. Overall, the decision for re-initiation should involve a detailed patient–physician centered conversation alongside risk factor modification.

## 5. Potential for Hepatitis B and C Reactivation

Chronic hepatitis B virus (HBV) and hepatitis C virus (HCV) infections have long been established as major risk factors for the development of cirrhosis and hepatocellular carcinoma (HCC). According to one study, 57% of cirrhosis was attributed to HBV (30%) or HCV (27%), and 78% of HCC was attributed to these infections (53% for HBV and 25% for HCV) [81]. There are several mechanisms through which HBV and HCV may lead to the progression of HCC. Similarities between both viruses include persistent liver inflammation, impaired antiviral immune response, oxidative stress, and viral-mediated deregulation of cellular signaling pathways [82]. Chronic infection with these viruses leads to an adaptive immune response involving CD8+ T cells and pro-inflammatory cytokines, resulting in liver inflammation and continuous necrosis of hepatocytes. The proliferation of hepatobiliary progenitor stem cells as a compensatory response to this ongoing inflammation favors the development of cellular transformation from genetic/epigenetic lesions, oncogenic mutations, and genomic instability [82,83]. Prolonged inflammation, which favors this type of remodeling of the liver microenvironment and extensive oxidative stress, can give way to tumorigenesis. HBV, more specifically, works towards genomic instability by integrating its DNA into the host genome. This allows for chromosomal instability, the creation of proto-oncogenes and tumor suppressors from insertional mutagenesis, and the expression of mutant HBV proteins such as HBsAg, HBcAg, and HBx proteins [84,85]. On the other hand, HCV causes metabolic reprogramming, resulting in steatosis. HCV core protein, through multiple mechanisms, is able to modify lipid metabolism—including the inhibition of LD motility, peroxisome proliferator-activating receptor-α/γ (PPAR-α/γ), lipid turnover, and export/degradation. The mitochondrial and endoplasmic reticulum oxidative stress, which results from enhanced lipogenesis, produces lipotoxicity that can promote carcinogenesis. Furthermore, HCV’s several proteins (including core, E2, NS2, NS3, NS4A, NS5A, and NS5B) play a large role in stimulating cell proliferation, growth, and survival pathways [82,86]. These include transforming growth factor β (TGF-β), nuclear factor κB (NF-κB), tumor necrosis factor α (TNF-α), cyclooxygenase-2 (COX-2), Wnt/β-catenin (WNT), and vascular endothelial growth factor (VEGF) [87].

HBV reactivation has been seen to occur in several treatments of HCC, including chemotherapy, biologics, and immunosuppressive medications. This indicates that suppressing the immune system with anti-cancer therapies is a key component of viral reactivation in these patients. Studies have shown that the degree and intensity of anti-cancer therapy is directly proportional to the risk of reactivation [88,89]. ICIs have shown promise as an alternative treatment option for HCC, but they can also lead to immune-related adverse events, including hepatitis viral reactivation. Previous studies have confirmed that HBV reactivation in patients with HCC leads to decreased survival given that it augments the HCC-related decline of hepatic function [89]. One specific study noted that of the 102 patients undergoing systemic chemotherapy for HCC, 58% of these patients developed hepatitis, 36% of which was secondary to HBV reactivation. The mortality rate of these patients who developed HBV reactivation was 30% [90]. This highlights the importance of gaining an understanding of the risk of HBV reactivation in patients with HCC undergoing ICI therapy.

Previously established risk factors for HBV reactivation in general are male sex, older age, cirrhosis, immunosuppressive medications, HBsAg positivity, HBeAg positivity, and higher baseline serum HBV DNA levels [91]. The risk of HBV reactivation in patients with HCC undergoing ICI treatment, however, is not well understood, given that these therapies have only recently surfaced as an option for these patients [92]. Furthermore, it has been previously established that positive HBsAg and treatment with immunosuppressives or chemotherapy are indicators for starting antiviral prophylaxis and monitoring for HBV reactivation; however, studies have only recently began looking at the effects of ICIs on reactivation [91]. In one particular systematic review and meta-analysis looking at hepatitis B reactivation in multiple cancer types, with all of them being treated with immune checkpoint inhibitors, results from a subgroup analysis showed that reactivation was more common in patients with HCC compared to other cancer types (1.9% (95% CI: 0–5.7%; I2  =  92.52%, *p*  <  0.001) vs. 0.5% (95% CI: 0–2.2%; I2  =  72.37%, *p*  <  0.001)). This study also showed that a HBsAg-positive status in general, across all cancer subtypes, was associated with a higher risk of reactivation (1.3% (95% CI: 0–4.5%; I2  =  87.44%, *p*  <  0.001) and 0 (95% CI: 0–0; I2  =  0, *p*  =  0.796)) [93]. These results corresponded with the results from another retrospective territory-wide cohort study looking at HBV reactivation in patients with cancer treated with immunotherapy. In this study, two patients who were HBsAg-positive had reactivation, and no patients that were HBsAg-negative had reactivation. It is important to note that, overall, reactivation was rare in this study and all patients included were on antivirals [94]. Furthermore, a historical cohort study of 3465 patients with ICI-treated cancer that included patients with HCC showed the incidence of reactivation was 0.14%, 1.0%, and 0.0% in all patients, HBsAg-positive patients, and HBsAg-negative patients, respectively. Regarding antiviral prophylaxis, rates of reactivation were 0.4% and 6.4% in patients with and without it, respectively [92]. In addition, a retrospective cohort study looking at HBV reactivation in patients with HBsAg-positive cancer being treated with anti-PD-1/PD-L1 antibody therapies showed that none of the antiviral prophylaxis therapies were the sole substantial risk factor for HBV reactivation (OR 17.50 [95% CI, 1.95–157.07]; *p*  = 0.004). This study concluded that HBsAg positivity does not necessitate a contraindication for ICI treatment in patients with cancer [95]. The CheckMate 040 study examined patients with HCC treated using nivolumab; of the 15 HBV-infected patients, none had HBV reactivation, but all of them were on appropriate antiviral prophylaxis and were subjected to regular monitoring for HBsAg [40]. In a retrospective prospective design of 62 patients with a chronic or prior HBV infection on nivolumab or pembrolizumab treatment for unresectable HCC, none of the patients with HBV DNA ≤ 100  IU/mL or HBV DNA > 100  IU/mL on prophylaxis had reactivation. Only one of the six patients not treated with prophylaxis had reactivation after 9 weeks of ICI therapy [96].

Altogether, the literature suggests that the risk of HBV reactivation in patients with HCC treated with ICIs is extremely low, but it is higher in HBsAg-positive patients compared to HBsAg-negative patients. However, even if HBsAg is present, the literature demonstrates that this is not a contraindication to ICI therapy and that even without antiviral prophylaxis in these patients, risk of reactivation is low. However, management practice should be tailored towards initiating antiviral prophylaxis in HBsAg-positive patients to prevent reactivation, irrespective of other factors, with monitoring of viral load throughout ICI therapy [88]. One particular aspect of management that could be seen in the near future is the use of novel, high-sensitivity biomarkers, given that, although low-risk, HBsAg-positive patients have a higher risk of HBV reactivation when under ICI therapy. According to one study, a fully automated high-sensitivity HBsAg assay by Lumipulse HBsAg-HQ has a 10-fold higher sensitivity than currently implemented conventional assays. In addition, highly sensitive HBsAg assays using a semi-automated immune complex transfer chemiluminescence enzyme immunoassay and a fully automated, high-sensitivity hepatitis B core-related antigen assay have also been developed. These tests have been shown to be comparable in terms of their HBV reactivation diagnostic capability compared to HBV’s viral load [97].

Although we have evidence that HBV reactivation can occur with ICI treatment, the exact mechanism that leads to this reactivation remains to be fully elucidated. One mechanism theorized is based on the concept of immune “exhaustion”, which is evidenced by weak or absent virus-specific T-cell reactivity in chronic hepatitis B [92]. Along with this, reduced cytokine production and an increased expression of inhibitory receptors including PD-1 and CTLA-4 have been seen [98]. Thus, it would be expected that the blockage of these axes would result in intensifying an immune result, which would decrease the risk of viral reactivation. However, it appears that a paradoxical effect occurs, which is suspected to be due to a disruption of immune homeostasis, resulting in viral reactivation rather than viral suppression [92,99]. Further studies will need to be performed to clarify and highlight these poorly understood mechanisms underlying HBV reactivation secondary to ICI treatment.

HCV reactivation in patients with HCC being treated with ICI therapy has been less studied, although a few studies have demonstrated a low risk of reactivation in this patient population. One clinical trial looking at the effects of tremelimumab in patients with HCC and chronic HCV infection revealed no risk of reactivation of HCV and actually demonstrated that these patients displayed a decrease in viral load and progressive continuation of this trend over a 3-month period [32]. Another single-institution retrospective chart review of patients with ICI-treated cancer with active or resolved HCV showed no evidence of worsening adverse events compared to patients without active or resolved HCV. In this study, hepatocellular carcinoma accounted for 30% of the cases included [100]. These findings are further substantiated by other multicenter studies, confirming the safety of ICI treatment in patients with HCC without the risk of HCV reactivation [101,102]. Overall, the history of HCC infection is not a contraindication to ICI therapy in these HCC patients, and there is a low concern for reactivation in this population [103,104].

In conclusion, hepatitis B and C reactivation are both legitimate concerns in patients with HCC treated with ICIs. However, overall, the risk appears to be low. Moreover, to reduce the risk as much as possible, close monitoring of viral markers and antiviral prophylaxis in high-risk patients can be used as ways to help mitigate the risk of reactivation and associated complications. Further research needs to be conducted to better understand the mechanisms underlying reactivation, delving deeper into the risks of HCV reactivation and identifying additional predictive biomarkers resulting in reactivation in this patient population.

## 6. Autoimmune Hepatitis Caused by ICIs

Autoimmune hepatitis (AIH) caused by ICIs is a rare but potentially severe adverse event in patients with cancer undergoing immunotherapy. AIH is an inflammatory liver disease characterized by the immune-mediated destruction of hepatocytes. It is typically treated with immunosuppressive therapy, such as corticosteroids [105]. The use of ICIs can trigger or exacerbate AIH by dysregulating the immune system and inducing an autoimmune response [71]. Several studies have reported cases of AIH in patients with cancer treated using ICIs. A study by De Martin et al. described 23 cases of AIH induced by ICIs, with most patients presenting with elevated liver enzymes and positive auto-antibodies [72]. Another study by Zen et al. reported on 48 cases of ICI-induced AIH, highlighting the importance of early recognition and prompt treatment to prevent liver damage [106].

The pathogenesis of AIH caused by ICIs is not fully understood, but is believed to involve a dysregulation of the immune system and the activation of autoreactive T cells [71]. ICIs work by blocking inhibitory signals in the immune system, enhancing the immune response against tumor cells. However, this immune activation can also lead to the development of immune-related adverse events (irAEs), including AIH [107]. In AIH, the immune system mistakenly targets the hepatocytes in the liver, resulting in inflammation and damage. The exact mechanism by which ICIs trigger or exacerbate AIH is not fully understood, but there are several proposed mechanisms:Activation of Autoreactive T Cells: ICIs can lead to the activation of autoreactive T cells, which recognize self-antigens present on hepatocytes as being foreign. This activation may occur be due to the breakdown of immune tolerance, where the immune system fails to recognize self-antigens as “self” and mounts an immune response against them.Dysregulation of Regulatory T Cells: Regulatory T cells are a subset of T cells that play a crucial role in maintaining immune tolerance and preventing autoimmune diseases. ICIs may disrupt the balance between effector T cells and Tregs, leading to an imbalance in the immune response and allowing for the development of AIH.Release of Pro-inflammatory Cytokines: ICIs can induce the release of pro-inflammatory cytokines, such as tumor necrosis factor-alpha (TNF-α) and interferon-gamma (IFN-γ), which contribute to the inflammation and damage observed in AIH. These cytokines can further perpetuate the autoimmune response and promote hepatocyte destruction.Genetic Predisposition: Genetic factors may also play a role in the development of AIH in response to ICIs. Certain genetic variations involved in immune regulation and inflammation have been associated with an increased risk of autoimmune diseases, including AIH. These genetic predispositions may contribute to the susceptibility of developing AIH following ICI therapy [71,105].

Understanding the pathogenesis of ICI-induced AIH is crucial for early recognition, appropriate management, and the development of strategies to minimize the risk of this adverse event [105]. Further research is needed to elucidate the exact mechanisms involved and to identify potential biomarkers for predicting and monitoring AIH in patients receiving ICIs. It is important to note that AIH can also occur spontaneously without the use of ICIs, and the underlying mechanisms may overlap with ICI-induced AIH. The interplay between ICIs and pre-existing genetic factors or autoimmune conditions may contribute to the development of AIH in susceptible individuals.

AIH and hepatotoxicity from ICIs can present with overlapping clinical presentations and laboratory findings, making their differentiations challenging. Certain distinguishing features such as auto-antibodies, histological findings, response to treatment and presence of other irAEs can help differentiate the two conditions. Patients with AIH induced by ICI therapy may present with nonspecific symptoms such as fatigue, malaise, abdominal discomfort, and jaundice. Laboratory findings often include elevated transaminases, hyperbilirubinemia, and elevated levels of serum immunoglobulins and auto-antibodies. The presence of specific auto-antibodies (such as anti-nuclear antibodies, anti-smooth muscle antibodies, and anti-microsomal antibodies) may support the diagnosis of AIH. In contrast, patients with hepatotoxicity may also present with nonspecific symptoms and transaminitis; however, hyperbilirubinemia and the presence of autoimmune markers are typically absent [108]. In cases where a liver biopsy is obtained, features of chronic inflammation and lymphoplasmacytic infiltrates will be present in cases of AIH secondary to ICI therapy [105]. Those with hepatotoxicity will often exhibit features of a drug-induced liver injury, such as centrilobular necrosis, eosinophilic infiltration, and cholestasis. Typically, patients respond well to immunosuppressive therapy in those with AIH compared to hepatotoxicity, where withdrawal of the offending agent (in this case, ICIs) often leads to the resolution of the symptoms that are present and can help further delineate the underlying pathology [109]. Patients with other irAEs affecting different organ systems, such as dermatitis, colitis, thyroiditis, or pneumonitis, may suggest that there is a drug-induced immune-mediated mechanism involved [108]. A multidisciplinary approach should be taken in such cases. 

To summarize, AIH caused by ICIs is a rare but potentially serious complication in patients with cancer undergoing immunotherapy. Early recognition, prompt treatment, and close monitoring are essential to prevent liver damage and optimize patient outcomes.

## 7. Risk of Gastrointestinal Bleeding with ICI Use

Gastrointestinal (GI) bleeding is a feared complication of liver disease. As previously discussed, cirrhosis is the major predisposing factor for the development of HCC. The process of fibrosis in the pathogenesis of cirrhosis causes a disruption of the blood flow between hepatocytes and sinusoids, which in turn impedes portal inflow, resulting in portal venous hypertension (PVT) [110]. PVT, defined as a pressure gradient between the hepatic and portal vein of 10mmHg or greater, promotes the development of collateral vessels that shunt portal blood to systemic veins, called varices. These varices most often occur in esophagogastric channels and can be prone to ruptures and hemorrhages [110]. Gastric varices (GVs) bleed less frequently than esophageal varices (EVs) and account for 10–30% of all variceal hemorrhages [111]. The annual bleeding risk in patients with cirrhosis varies according to each patient’s liver function; patients with decompensated cirrhosis and large EVs have a 42% to 76% 1-year risk of bleeding [112]. For patients with HCC, the prevalence of EVs is greater than 50%. In these patients, the rate of death from hemorrhages was 12.3%, compared to 5.5% for those without EVs. The presence of EVs, independent of the HCC stage, increased the risk of death by 25–28% [113]. Given this risk, it is imperative to consider the risk profile of each chemotherapeutic agent prior to initiating treatment in these patients. 

Based on the current literature, when used as a monotherapy, ICIs themselves do not increase the risk of GI bleeding [32,40,44,114,115,116]. It is important to note, however, that ICIs are currently only approved for use in patients with HCC in combination with other agents [117,118]. 

The IMBrave 150 trial established the ICI/anti-vascular endothelial growth factor combination atezolizumab/bevacizumab (atezo-bev) as the standard-of-care treatment for advanced HCC [46,118,119,120]. While atezolizumab itself has not been implicated with an increased risk of bleeding in other approved cancer treatments (such as in non-small cell lung cancer), it is only approved for HCC in combination with bevacizumab, which has been found to increase the risk of bleeding [121]. Bevacizumab is a monoclonal antibody that selectively binds to the vascular endothelial growth factor (VEGF). The physiological role of VEGF is multifaceted, but is generally involved in the function and survival of blood vessels. The disruption of this function decreases patients’ regenerative capacity, providing therapeutic value in the treatment of certain malignancies, but it can also predispose patients to bleeding. The prior standard-of-care treatment in patients with advanced HCC—tyrosine kinase inhibitors, i.e., sorafenib, lenvatinib—affect multiple tyrosine kinase receptors involved in angiogenesis and cell proliferation. These include the VEGF receptor, thereby increasing the bleeding risk through a similar mechanism [122]. 

A 2023 meta-analysis evaluating the safety and efficacy of atezolizumab/bevacizumab in patients with advanced HCC found that the cumulative incidence of variceal bleeding was 4.7% across 16 studies and is comparable to the variceal bleeding risk posed by tyrosine kinase inhibitors [123]. Of the ten studies included that reported conducting screening protocols for varices, 1130/1429 (79%) were screened prior to therapy, and 46% reported the presence of EVs and GVs. The incidence of EV bleeding (EVB) was similar between those who received prophylactic therapy (esophageal variceal ligation or NSBB) versus those who did not. The meta-analysis identified the presence of a portal vein thrombus (PVT) as a risk factor for EVB [123]. Other studies that specifically evaluated EVB in advanced HCC patients receiving atezo-bev identified the presence of high-grade EVs (grade 2 or grade 3) and prior EVB as additional risk factors [124,125]. A large multicenter, retrospective study of patients receiving lenvatinib determined that the risk of EGV bleeding was limited to patients with platelets < 150,000, less preserved liver function, and with baseline neoplastic PVT. While this study does not provide clear guidance for atezo-bev use, considering these factors can help guide risk stratification [126].

The AASLD recommends baseline endoscopic screening for EVs and GVs prior to initiating therapy with atezo-bev [120]. One study suggests treating EVs with banding and/or medical therapy prior to initiating atezo-bev therapy effectively controls patients’ risk of bleeding, but the optimal treatment regimen prior to treatment is unknown [120,127]. Further studies are needed to evaluate the optimal timing of baseline EGD and surveillance of EVs prior to and during systemic anti-cancer therapy.

In 2023, the FDA approved a second first-line ICI combination regimen for the treatment of advanced HCC, called STRIDE, in response to the HIMALAYA trial [35]. STRIDE consists of tremilimumab and durvalumab; this combination does not increase patients’ risk of bleeding. Therefore, in patients at high risk for EGV bleeding, STRIDE should be considered [120]. Studies comparing STRIDE to other first-line therapies are needed to further evaluate its safety and efficacy, and optimal treatment regimens.

In conclusion, variceal bleeding is a significant risk in patients with advanced HCC. Despite ongoing trials evaluating combinations of ICIs, atezo-bev remains the first choice when choosing a first-line agent [117,118,120]. PVT, prior history of EVB, and grade 2 or 3 EVs increase the risk of variceal bleeding in patients undergoing atezo-bev treatment [123,124,125]. Baseline screening EGD is recommended prior to starting atezo-bev therapy, and adequate treatment of EVs with banding and/or medical management prior to initiating treatment may serve as a control for preventing an increased risk of bleeding [127]. For patients with an unacceptable bleeding risk, practitioners and patients should consider STRIDE therapy as a first-line agent [120].

Future studies are needed to evaluate this topic further, especially as more ICIs/combinations of ICIs make their way onto the market.

## 8. Conclusions and Future Directions

ICIs have revolutionized the playing field in the world of treatment options for patients with advanced HCC. Although they have been on the market for a while, the results from previous trials and studies have been encouraging. However, as highlighted in this paper, ICIs still pose risks for patients, especially regarding the liver.

While, presently, there are studies and guidelines present for clinicians that can help in the management of ICH, a challenging component of this is the fact that these guidelines do not necessarily consider that the patient may have an underlying liver dysfunction at baseline, which is something that is commonly seen in patients with advanced HCC. Thus, ICI-related effects on the liver may be magnified in patients with advanced HCC.

There are only a few studies currently investigating the incidence of ICH, specifically in patients with advanced HCC; however, as ICIs become more commonplace in advanced HCC, further studies should be conducted to investigate hepatic-related side effects secondary to these agents. Furthermore, with the increased use of these agents (and with newer studies conducted) in patients with advanced HCC, we may be able to further understand the risk, incidence, and mechanism of ICH, hepatitis viral reactivation, AIH, and other hepatic-related side effects in this patient population as well as those who may be most susceptible to these hepatic-related side effects.

Regardless of the potential side effects of these agents, this is an exciting time for being in this field. With the continued evolution of this field and the increase in the use of these agents, physicians will become more familiar with both their benefits and risks.

## Figures and Tables

**Table 1 cancers-16-02042-t001:** Grading of hepatotoxicity using the drug-induced liver injury (DILI) network and Common Terminology Criteria for Adverse Events (CTCAE).

Grade	Drug-Induced Liver Injury (DILI) Network	Common Terminology Criteria for Adverse Events (CTCAE)
1	AST and/or ALP levels are elevated, total serum bilirubin is <2.5 mg/dL, and there is no coagulopathy (INR < 1.5)	AST and/or ALT < 3 times the ULN and/or total bilirubin < 1.5 times the ULN
2	AST and/or ALP levels are elevated, total serum bilirubin is ≥2.5 mg/dL, or coagulopathy is present (INR ≥ 1.5) without elevated bilirubin levels	AST and/or ALT levels are 3 to 5 times the ULN and/or total bilirubin is 1.5 to 3 times the ULN
3	AST and/or ALP levels are elevated, total serum bilirubin is ≥2.5 mg/dL, and there is prolonged hospitalization due to drug-induced liver injury	AST and/or ALT levels are 5 to 20 times the ULN and/or total bilirubin is 3 to 10 times the ULN
4	AST and/or ALP levels are elevated, total serum bilirubin is ≥2.5 mg/dL, and the patient is demonstrating signs of hepatic decompensation (ascites, encephalopathy, INR ≥ 1.5) or other organ failure	Hepatic decompensation (ascites, encephalopathy, INR ≥ 1.5), AST and/or ALT levels > 20 times the ULN and/or total bilirubin > 10 times the ULN
5	Death or liver transplantation needed for survival	Death

AST: aspartate aminotransferase; ALT: alanine aminotransferase; ALP: alkaline phosphatase; INR: international normalized ratio; ULN: upper limit of normal.

**Table 2 cancers-16-02042-t002:** American Society of Clinical Oncology management of immunotherapy-induced hepatotoxicity.

Inform patients to contact their health care provider if they experience any of the following symptoms: jaundice, nausea and/or vomiting, right-upper quadrant abdominal pain, decreased appetite, increased bruising or bleeding, darkening of the urine, increased drowsiness. Once diagnosed with ICH, discontinue all hepatotoxic agents and unnecessary medications.	
Grade 1	- Supportive care and symptom control. - Can continue ICI treatment, but has to have close monitoring of liver tests and function 1 to 2 times weekly.
Grade 2	- Suspend ICI treatment. - Monitor liver tests every 3 days; if they are persistently elevated and the patient is symptomatic, start corticosteroids (prednisone 0.5–1 mg/kg/day or equivalent). - Avoid infliximab. - Can consider resumption of the ICI followed by a taper of the steroids only when symptoms improve to grade 1 or less on prednisone 10 mg/day. Taper the steroids over at least 1 month.
Grade 3	- Stop the ICI permanently. - Initiate methylprednisolone 1–2 mg/kg/day or equivalent and monitor liver tests daily or every other day. - If there is no improvement after 3 days, consider mycophenolate mofetil or azathioprine. - Refer to hepatology. - Consider hospitalizing the patient if their AST/ALT ratio > 8 times the ULN and/or total bilirubin is 3 times the ULN. - Avoid infliximab. - The optimal duration of the treatment is not known; steroid taper should be trialed over 4 to 6 weeks if symptoms improve to grade 1 or less; increase the dose if needed.
Grade 4	- Stop the ICI permanently. - Initiate methylprednisolone 2 mg/kg/day or equivalent and monitor liver tests daily. - If there is no improvement after 3 days, consider mycophenolate mofetil. - Refer to hepatology. - Hospitalize the patient. - Avoid infliximab. - The optimal duration of the treatment is not known; steroid taper should be trialed over 4 to 6 weeks if symptoms improve to grade 1 or less; increase the dose if needed.

AST: aspartate aminotransferase; ALT: alanine aminotransferase; ULN: upper limit of normal.

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
