# Peer review of "Immune Checkpoint Inhibitors in Hepatocellular Carcinoma and Their Hepatic-Related Side Effects: A Review"

_cancers, 2024, doi:10.3390/cancers16112042_

Round 1
Reviewer 1 Report
Comments and Suggestions for Authors
I would add some graphs regarding OFS and or OS of the main Cited papers
Comments on the Quality of English LanguageThe manuscript need minor English revisions
Author Response
Thank you very much for taking the time to review our manuscript. Please find our responses below and the corresponding revisions highlighted in the re-submitted file.
Comment #1: I would add some graphs regarding OFS and or OS of the main Cited papers
Response #1: Thank you for this comment. We did not know what was being referred to with the abbreviations 'OFS' and/or 'OS'. We reached out a couple of times to the assistant editor to try to get clarification on this so we could address your comment, however, our assistant editor never heard back in time before the revision due date.
Comment #2: The manuscript need minor English revisions
Response #2: Thank you for highlighting this; we have reviewed the manuscript and made some minor grammatical adjustments.
Reviewer 2 Report
Comments and Suggestions for Authors
This review serves as a guide for clinicians on ICI’s role in the management of HCC and their potential side effect on ICI-related hepatotoxicity. This review provides a safety reference for the application of ICI in the treatment of HCC.
The side effects of ICI treatment in HCC are not only liver toxicity, but need to be considered systemically. For example, systemic T-cell exhaustion causes liver toxicity. HCC patients themselves have liver function damage, how to distinguish whether the liver damage is caused by ICI alone or synergistically increases the liver damage. The author needs to discuss the above questions in the discussion section.
Comments on the Quality of English LanguageThe English language is generally smooth.
Author Response
Thank you very much for taking the time to review our manuscript; please find the detailed responses below and the corresponding revisions/corrections highlighted in the re-submitted file.
Comment #1: The side effects of ICI treatment in HCC are not only liver toxicity, but need to be considered systemically. For example, systemic T-cell exhaustion causes liver toxicity.
Response #1: Thank you for pointing this out. We briefly mentioned other ICI-related adverse events (outside hepatotoxicity) in the introduction (including pneumonitis, myocarditis, colitis, etc.) but chose not to delve into the other systemic effects of ICIs in HCC throughout the rest of the paper since we wanted to focus primarily on liver-related adverse effects. We have added a small excerpt in regard to T cell exhaustion and its association with liver damage.
Comment #2: HCC patients themselves have liver function damage, how to distinguish whether the liver damage is caused by ICI alone or synergistically increases the liver damage.
Response #2: Thank you for this comment. It is indeed difficult in some cases to distinguish whether hepatotoxicity is caused by ICIs alone or acts synergistically. The diagnosis can be challenging. Based on our review, ICI-induced hepatotoxicity is primarily a diagnosis of exclusion and utilizing scoring systems (such as RUCAM scoring system) along with history taking can help to establish causality. See lines 342 through 371 in the manuscript. Clinicians can have a good idea of whether ICIs are directly causing hepatotoxicity by taking into account the time to onset of elevated LFTs after initiation of ICIs, and then monitoring of LFTs after cessation of the ICI.
Reviewer 3 Report
Comments and Suggestions for Authors
This manuscript describes immune checkpoint inhibitors (ICI), and lists examples of, their success rate in hepatocellular carcinoma (HCC), and hepatic-related side effects.
The authors primarily address examples of anti-CTLA-4 and anti-PD-1 monoclonal antibodies approved for use in HCC treatment.
The authors discuss sixteen studies from 2013 through to the present.
It is important to discuss available treatments and their success rate, and their side effects, and therefore this piece is timely.
Author Response
Thank you very much for taking the time to review our manuscript; we are glad that you found this piece to be timely and we hope that it will serve as a useful guide to clinicians in the future.